# Erosion-Transportation Processes Influenced by Spatial Distribution of Terraces in Watershed in the Loess Hilly–Gully Region (LHGR), China

Zhe Gao [1], Genguang Zhang [1,*], Henghui Fan [1], Qianqian Ji [2,3,4], Anbin Li [1], Yuanyuan Zhang [3,5], Boyan Feng [1], Yuanhao Yu [1], Lin Ma [1] and Jianen Gao [1,2,3,5,*]

1   College of Water Resources and Architectural Engineering, Northwest A&F University, Yangling, Xianyang 712100, China; gaozhe19900108@163.com (Z.G.); fanhenghui@hotmail.com (H.F.); adrianlab@foxmail.com (A.L.); fby941002@163.com (B.F.); y15559636359@126.com (Y.Y.); m18856335765@163.com (L.M.)
2   Institute of Soil and Water Conservation, Northwest A&F University, Yangling, Xianyang 712100, China; jqq029@163.com
3   Research Center on Soil & Water Conservation, Ministry of Water Resources, Yangling, Xianyang 712100, China; yuanyuan2567@163.com
4   School of Civil Engineering, Yangling Vocational & Technical College, Yangling, Xianyang 712100, China
5   Institute of Soil and Water Conservation, Chinese Academy of Sciences and Ministry of Water Resources, Yangling, Xianyang 712100, China
*   Correspondence: zgg64@163.com (G.Z.); gaojianen@126.com (J.G.)

**Abstract:** How to optimize the spatial distribution of terraces in the watershed is an important scientific problem. It was researched through a watershed solid-scale physical model based on the 3D reappearance of a scene under the Cartesian coordinate system, with the lowest point of the watershed as the origin. The results showed that the change of the spatial pattern of terraced fields in the basin had an important impact on the production of runoff and sediment. There was an approximate quadratic-function relationship between the spatial location and the parameters of runoff and confluence. If $R_t$ was terrace-erosion-reduction benefit, it could be defined as the reduction in the watershed-erosion modulus per unit of terrace area. The longitudinal distribution of $R_t$ was upper and middle > lower parts, and the vertical distribution of $R_t$ was high > low place. The erosion reduction was 77.67% of the terraces of the middle and upper, occupying 33% of the watershed area. The change of the $R_t$ was logarithmically related to the relative distance ($r$) from the center of the terrace. When $r$ was around 0.35, there was an inflection point in $R_t$ growth. The results of this study have important practical significance for the planning and construction of terraces in the watershed.

**Keywords:** erosion transport; terraced fields; spatial distribution; Loess hilly–gully region

## 1. Introduction

A terrace is one of the important soil- and water-conservation measures [1]. It has a wide range of applications in the world. The Loess Plateau is the region with the most serious soil erosion in the world [2,3]. For soil-erosion control and high-quality development [4,5], a large number of terraced fields will be built on the Loess Plateau. It was of great practical significance to study the influence of the spatial-pattern optimization of terraced fields on erosion and transport in the basin.

Many scholars have studied the erosion-transportation processes and the influence by the spatial distribution of terraces in the watershed [1,6–10]. Feng et al. [11] thought the runoff process on the slope surface was affected by the type of rainfall intensity, the structure of terraced fields, the agricultural-planting pattern, and other factors in the watershed. Liu et al. [12] proposed a new hybrid vector-grid method to optimize the regional water system and build terraces as well as other soil- and water-conservation measures. The

results show that, whether in geometric form or in spatial topology, and this method can obtain a more realistic watershed network than the traditional method. Gao, Shao, and Liang et al. [13–15] used the terrace module of the soil- and water-assessment tool (SWAT model) and the water-erosion-prediction project (WEPP model) to optimize the structure of the terraced fields, based on the regulation of crop-water demand and rainfall runoff. Antle et al. [16] considered that the layout of terraced fields did not affect the characteristic values of soil stability, bulk density, and water permeability in the basin. Chen [17] conducted a systematic analysis of 46 pieces of research from the literature on water and sediment reduction of terraced fields, and concluded that 48.9% and 53.0% were, respectively, the average water and sediment reduction effects of six types of terraces in China. Zhu [18] found that the role of vegetation and terraces in reducing peak flow was reduced, with rainfall-return periods greater than 20 years. Liu, Wang, and Gao et al. [19–22] introduced the concept of terrace ratio and sand-reduction range, proposed a calculation method of sediment reduction by combining satellite-remote sensing with field investigation, analyzed the changes of forest and grass vegetation coverage (from 2010 to 2013), constructed the relationship between the different scale and level terraces as well as sediment-reduction range, and gave the layout threshold of terraces in the watershed.

The above research and achievements mainly focused on mathematical simulation and data analysis. The mathematical model was a scenario simulation based on the verification of water and sediment. Qualitative analysis was possible, but it required a lot of demonstrations. Due to the three-dimensional nature of the actual watershed and the complexity of the underlying surface, it was impossible to accurately describe the mechanism and process of the impact of terrace spatial distribution on water and sediment via mathematics. Therefore, it was necessary to explore a new 3D-solid-scale scene-reconstruction-simulation technology, to study its spatial-temporal-distribution influence.

This study conducted unique experiments by constructing a watershed-entity scale model, based on the 3D scenario representation of surface rainfall-runoff-erosion dynamics and similarity theory. The objectives were for investigating erosion-transportation processes, as influenced by the spatial distribution of terraces, giving their relations, and serving the terraced-field planning on Highly Managed Small Watersheds (HMSW).

## 2. Materials and Methods

### 2.1. Study Area

The Yan'gou watershed (36°21′–36°22′ N, 109°20′–109°35′ E) is located in the southern suburbs of Yan'an city in the middle of the LHGR (as shown in Figure 1), with an area of about 48 km$^2$ and an annual rainfall of 575 mm. The Kangjiagelao small watershed is located in the Yan'gou watershed, covering an area of 0.35 km$^2$, with a height of 189.7 m, and having a silt dam with a length of 11 m and a width of 5 m at the outlet of the gully. The terraced farmlands and terraced orchards in the watershed account for 15–20% of the total watershed area, and the remaining area contains arbores, shrubs, and herbaceous vegetation. The watershed vegetation coverage is 50~70%.

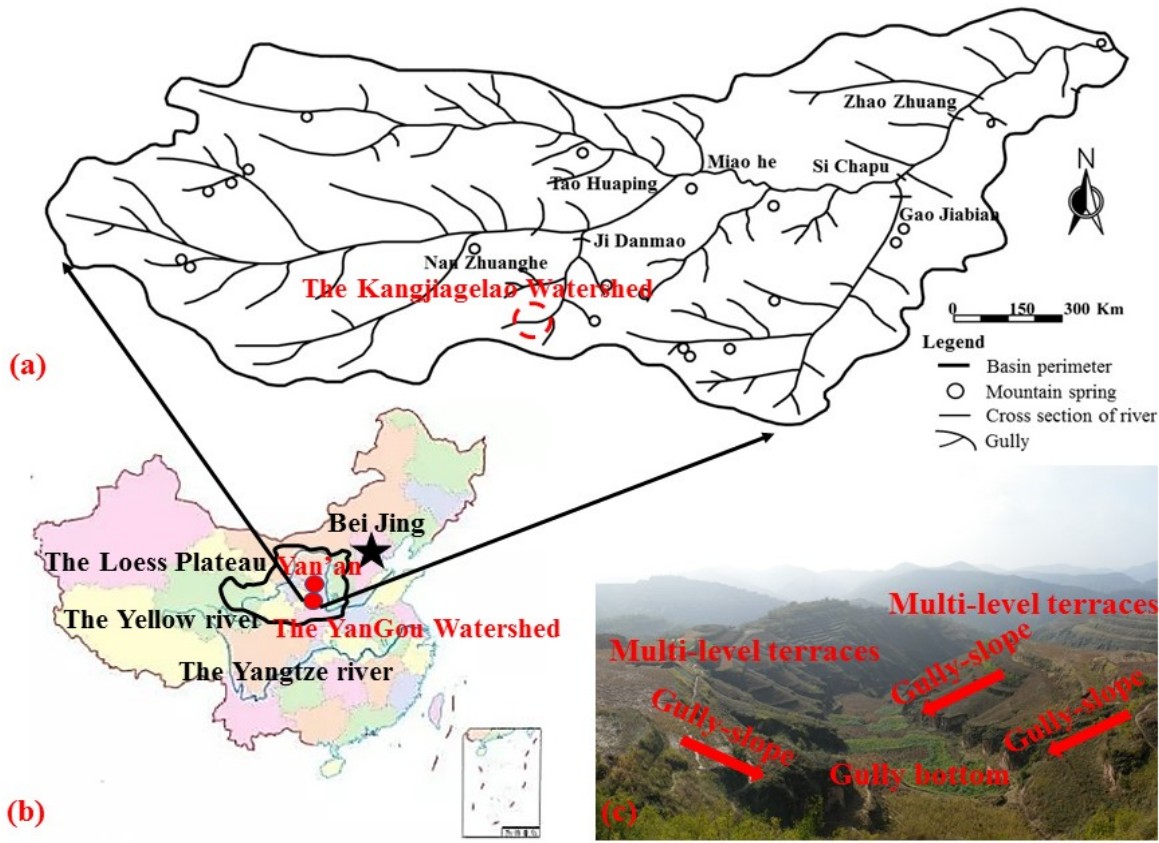

**Figure 1.** Location and different stages of the Kangjiagelao watershed; (**a**,**b**) location and early status of the Kangjiagelao watershed, (**c**) current status of the Kangjiagelao watershed.

*2.2. Model Principle and Scale Design*

2.2.1. Experimental Design

Designed Model:

Under the normality condition, the 1:100 (model:prototype) physical model of the Kangjiagelao watershed was designed based on the hydrodynamic principles of rainfall, runoff, sediment transport, infiltration, and similarity theory as well as the watershed-topographic map [23–25]. The model area was 34.17 m², the maximum length was 9.03 m, the maximum width was 7.23 m, and the maximum height difference was 1.89 m. The model was filled with sand and gravel, with plain soil and test soil from the bottom to the top, and all kinds of engineering measures were arranged, in full accordance with the prototype watershed (as shown in Figure 2a).

Designed Rainfall:

In this study, the same rainfall intensity was used (1.114 mm/min). Artificial rainfall was adopted for the rainfall simulation of the model. The rainfall equipment was a BX-1 portable field-rainfall device, and three groups were evenly arranged around the small watershed model. The rainfall intensity and soil water content needed to be re-calibrated before the formal test began. The rainfall uniformity taken during the test was more than 80%, which was expressed by the uniformity coefficient (*p*). These experiments were carried out under the conditions of the terrace layout, based on the relative horizontal distance, proportion of laying area, and laying height (as shown in Figure 2b,c). A spatial-coordinate system was established, with the lowest point of the watershed-outlet section as the origin o, the right helix of 90 degrees in the direction of the water flow as the *x*-axis direction, the reverse direction of the water flow as the *y*-axis direction, and the vertical direction of the x–y plane as the *z*-axis direction (as shown in Figure 2b,c). The relative distance was the ratio of the coordinates of the geometric center of the terrace to the corresponding

coordinates of the farthest and highest point in the watershed. The relative-layout-area ratio was the ratio of the total area of the terraces to the catchment area. There were 25 orthogonal design experiments with different terrace-area ratios and different positions under the same rainfall conditions (as shown in Table 1).

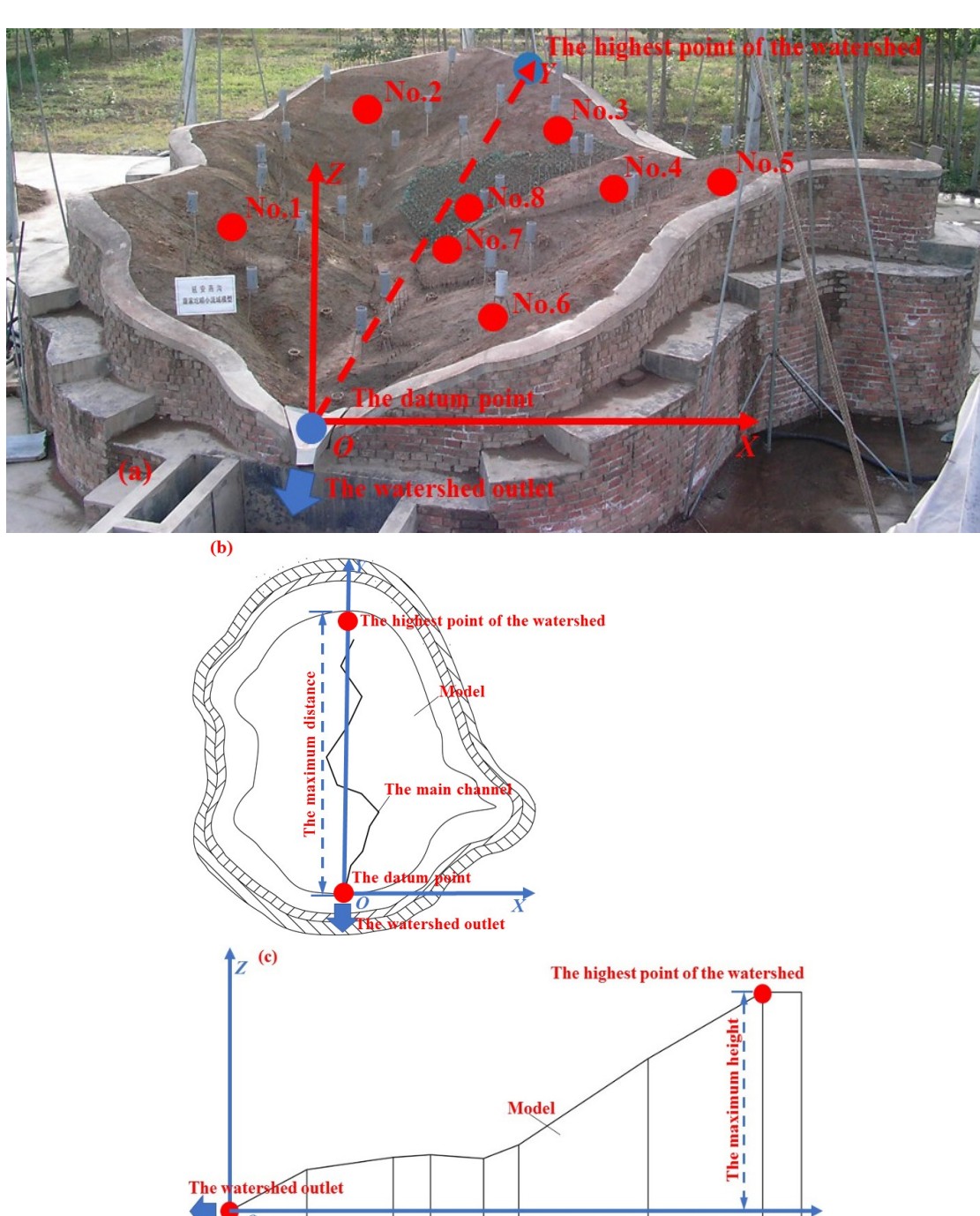

**Figure 2.** The model built according to the two situations; (**a**) no vegetation restoration, (**b**) longitudinal diagram of experimental design, (**c**) sectional drawing of experimental design.

**Table 1.** Working condition design of spatial distribution of terraces.

| Parameters | Value | | | | | Note |
|---|---|---|---|---|---|---|
| Distance | Up, Middle, Bottom (proportion of terrace area: 33.3%) | | | | | |
| Proportion of layout area (%) | 0 | 10 | 20 | 40 | 50 | Vegetation coverage: 70% |
| Height (m) | 30 | 50 | 80 | 100 | 150 | |
| Rainfall intensity (mm/min) | 1.14 | | | | | |

Designed Test Monitoring:

Three observation breakpoints were set on the slope, according to the flow direction of the basin (left and right slope, top of hills). The main channel was equidistantly 110 m from the upstream and the downstream, each with 1 observation section, for a total of 5 sections. Each observation section was measured for indicators, such as water-flow velocity and runoff depth, according to 3 lateral measuring points (left side, middle, and right side).

Designed Other Working Conditions (Terraces, Check Dam, and Vegetation):

According to the geometric scale of 1:100 of this model, the vegetation, terraced fields and check dam were arranged in an equal scale, based on the topographic and geomorphological features of the Kangjiagelao small watershed prototype, in 2003 and 2018. The value of the vegetation measure was 7 m and 15 m, after referring to the on-site vegetation height of the watershed. The terrace measures were scaled according to the original watershed layout, with a total of 8 and a total area of 0.163 m$^2$, accounting for about 47.8% of the total watershed area. The layout parameters of related measures were shown in Tables 2 and 3. The overall soil bulk density of the model was controlled at 1.35~1.75 g/cm$^3$.

**Table 2.** Design-foundation parameters of check dam.

| Dam Height (m) | Top Width (m) | Bottom Width (m) | Upstream Slope Ratio | Downstream Slope Ratio | Land Reclamation Ratio (%) |
|---|---|---|---|---|---|
| 30 | 5 | 55 | 1:1.5 | 1:1.5 | 5 |

**Table 3.** Foundation parameters of multistage terrace.

| Number | 1 | 2 | 3 | 4 | 5 | 6 | 7 | 8 |
|---|---|---|---|---|---|---|---|---|
| Area (km$^2$) | 0.0518 | 0.0123 | 0.0267 | 0.0175 | 0.0155 | 0.0330 | 0.0047 | 0.0010 |
| Proportion of area (%) | 15.2 | 3.6 | 7.8 | 5.1 | 4.5 | 9.7 | 1.4 | 0.3 |

2.2.2. Data Collection

The observation items of this study, mainly, included rainfall intensity, flow velocity, sediment content, soil water content, etc. (as shown in Table 4).

**Table 4.** Monitoring method of each monitoring index.

| Item | Monitoring Method | Meaning of Sign | Note |
|---|---|---|---|
| Rainfall intensity | $p = 1 - \frac{\sum_{i=1}^{n}|I_i - \bar{I}|}{n\bar{I}}$ | $p$ is uniformity index (%); $n$ is sample size; $I_i$ is rainfall intensity of the $i$th rain gauge (mm/min); $\bar{I}$ is average rainfall intensity of the whole sample (mm/min). | Rain intensity needs to be re-measured before each test. |
| Flow velocity, rate of flow, runoff depth | (1) Potassium permanganate dye tracking method, electrolytic salt particle tracing method; (2) $h = \frac{q}{VB}$ | $h$ is the runoff depth (cm); $V$ is the flow velocity (m/s); $B$ is breadth of water surface (cm); $q$ is discharge per unit width (m$^3$/s). | — |
| Sediment content | Oven-drying method | — | — |
| Soil water content | Alcohol burning method | — | Before the test, the soil water content was controlled at about 18%. |
| Soil bulk density | Ring knife method | The average volume soil bulk density of the watershed shall be controlled at about 1.30 g/cm$^3$. | — |
| Sand selected by experiments | The sediment incipient similarity was mainly considered in the test process, among them, the median particle size of soil is used as the representative index of particle gradation similarity. | The median particle size of the sample sand selected in this experiment is 0.009 mm. | — |
| Caking power | $N = \varphi \frac{\pi}{2} \rho \varepsilon_k d$ | $\varphi = \frac{1}{16}$, $\varphi$ is correction factor; $\rho$ is bulk density of water; $\varepsilon_k = 2.56$ cm$^3$/s$^2$. | When $d \geq 0.002$ mm, the bond force $N \geq 0.00002$ g·cm$^3$/s$^2$. |

*2.3. Determination of Controlling Water-Erosion Scale and Verification Based on Spatial-Distribution Similarity*

2.3.1. A Watershed Solid-Scale Physical-Model Principle and Scale Design

The concept of terrace-erosion-reduction benefit was introduced, in order to evaluate the influence of terrace layout in different spatial distributions on the changes of water and sediment in the watershed. It refers to the reduction of the erosion-transport modulus per unit of terrace area, before and after the construction of terraces in the watershed ($R_t$). It was calculated using Equation (1):

$$R_t = \frac{\Delta M_s}{F_T} \tag{1}$$

where $R_t$ is the terrace-erosion-reduction benefit, $\Delta M_s$ is the reduction of the erosion-transport modulus, before and after the construction of terraces in the watershed ($\Delta M_s = M_s - M_{si}$) (t/(km$^2$·a)), and $F_T$ is the layout area of the terraces (km$^2$).

Under the condition of a normal-scale model, for the watershed prototype, the above formula could be converted into:

$$R_{ty}F_{Ty} = M_{0y} - M_{iy} \tag{2}$$

The corresponding model equation is

$$R_{tm}F_{Tm} = M_{0m} - M_{im} \tag{3}$$

Under normal conditions, when the prototype was similar to the model, the prototype parameters were calculated

$$R_{ty} = \lambda_{R_t} R_{tm}, \; F_{ty} = \lambda_{F_t} F_{tm}, \; M_{0y} = \lambda_M M_{0m}, M_{iy} = \lambda_M M_{im}$$

where $\lambda_{R_t}$, $\lambda_{F_t}$ and $\lambda_M$ are the scales of terrace-erosion-reduction benefits, area, and erosion modulus, respectively. So, Equation (2) was written as

$$\frac{\lambda_F \lambda_{R_t}}{\lambda_M} R_{tm} F_{Tm} = M_{0m} - M_{im} \tag{4}$$

According to the similarity theorem, the prototype and the model must obey the same equation. So, there was

$$
\begin{gathered}
\frac{\lambda_F \lambda_{R_t}}{\lambda_M} = 1 \\
\lambda_{R_t} = \frac{\lambda_M}{\lambda_F} = \frac{\lambda_Q \lambda_s \lambda_t}{\lambda_F^2} = \frac{\lambda_l^{5/2} \lambda_s \lambda_l^{1/2}}{\lambda_l^4} = \frac{\lambda_s}{\lambda_l} \\
\lambda_{R_t} = \frac{\lambda_s}{\lambda_l}
\end{gathered}
\tag{5}
$$

where $\lambda_{R_t}$ is the scale of the terrace-erosion-reduction benefit, $\lambda_F$ is the scale of area ($\lambda_F = \lambda_l^2$), $\lambda_M$ is the scale of the erosion modulus ($\lambda_M = \lambda_Q \lambda_s \lambda_t / \lambda_F \lambda_M = \lambda_Q \lambda_s \lambda_t / \lambda_F$), and $\lambda_l$, $\lambda_Q$, $\lambda_S$, and $\lambda_t$ are the length, flow, sediment content, and time scale, respectively.

The scales of rainfall, water flow, sediment transport, and soil water were all determined with reference to previous research results [11,26,27] (as shown in Table 5).

**Table 5.** Summary of the main scales of the small watershed model.

| Name | | Scale Symbol | Scale Value | Remarks |
|---|---|---|---|---|
| Geometric similarity | Plane scale | $\lambda_l$ | 100 | Set |
| | Vertical scale | $\lambda_h$ | 100 | Set |
| | Vegetation-coverage scale | $\lambda_{fc}$ | 1 | Set |
| Rainfall similarity | Rain-intensity scale | $\lambda_i = \lambda_l^{1/2}$ | 10 | Derived |
| | Rainfall-capacity scale | $\lambda_P = \lambda_i \lambda_{t1}$ | 33.3 | Derived |
| | Rainfall-time scale | $\lambda_{t1}$ | $\lambda_{t1} \approx \lambda_{t'} = 3.33$ | Supposed |
| Water-flow similarity | Flow-rate scale | $\lambda_v = \lambda_l^{1/2}$ | 10 | Derived |
| | Flow-amount scale | $\lambda_Q = \lambda_l^{5/2}$ | 100,000 | Derived |
| | Roughness scale | $\lambda_n = \lambda_l^{1/6}$ | 2.15 | Derived |
| | Water-flow-time scale | $\lambda_{t1} = \lambda_l / \lambda_v = \lambda_l^{1/2}$ | 10 | Derived |

**Table 5.** *Cont.*

| | Name | Scale Symbol | Scale Value | Remarks |
|---|---|---|---|---|
| | Suspension-movement similarity | $\lambda_d = \lambda_l^{1/4}\lambda_v^{1/2}/\lambda_{(d_s-d)/d}^{1/2}$ | 3.16 | Derived |
| | Starting similarity | $\lambda_{u_c} = \lambda_u = \lambda_l^{1/2}$ | 10 | Derived |
| Erosion and sediment-movement similarity | Sediment-content scale | $\lambda_S$ | 3 | Calibrated and measured |
| | The similarity in the bed-surface-deformation time | $\lambda_{t'} = (\lambda_{\gamma'}/\lambda_S)\lambda_{t1}$ | $\lambda_{t'} = 10/3 = 3.3$ | Derived |
| | Sediment-transport-ratio scale | $\lambda_{Gs} = \lambda_Q\lambda_S$ | 300,000 | Derived |
| Soil water similarity | Soil water content scale | $\lambda_\theta$ | 1 | Derived |

### 2.3.2. Model Validation

Verification of Water and Sediment-Transport Similarity

After the terrain was repaired, verification tests were carried out under the condition of bare ground, with a rainfall intensity of 1.14 mm/min and erosion-equivalent rainfall of 140~150 mm. [23–25]. The process of rainfall, water, and sediment transport was further obtained (as shown in Table 6). Table 6 indicate that the confluence time, average velocity, maximum confluence, and annual erosion were very close to the data of the bare-slope-preparation test, after the initial construction of the model in 2003. The change process of flow and sediment concentration can reflect the change process of hydraulic erosion, in the corresponding prototype watershed.

**Table 6.** Summary of the main scales of the small watershed model.

| Item | Rain Intensity (mm/min) | Rainfall Capacity (mm) | Average Velocity (m/s) | Maximum Confluence (m³/s) | Confluence Time (h) | Annual Erosion (t/a) |
|---|---|---|---|---|---|---|
| Model test (2003) [25] | 1.14 | 170 | 0.84 | 5~6.39 | 0.3 | 2900 |
| Prototype (2003) [25] | 1.14 | 140~150 | 1.25 | 5~6.30 | 0.2 | 2800~3100 |
| This study | 1.16 | 232 | 0.96 | 5.58 | 0.35 | 2334 |

Verification of Erosion-Sediment-Yield Gradation

Figure 3 shows the sediment curves of the prototype and model. The model $D_{50}$ parameter was set to 0.028 mm, which is near the value of 0.026 mm used in the prototype. The soil selection of the model met the requirements.

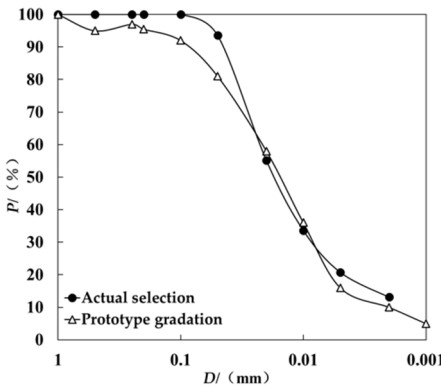

**Figure 3.** The sediment-particle-grading curves of the model and prototype.

In summary, the model satisfies the similarity of geometric, rainfall, flow, erosion-production-sediment transport, bed-surface deformation, etc. The changes in rainfall, flow, production sediment, and bed deformation are consistent with the prototype.

## 3. Results

### 3.1. Effect of Spatial Variation of the Same Terrace Area on Runoff and Sediment Yield in the Watershed

In order to study the effects of spatial changes of the same terrace area on erosion transport, runoff, and sediment yield in the watershed, the terraces were arranged in the upper, middle, and lower parts (*y* direction), according to about 20% of the area in this model. Figure 4 shows the change process of flow and sediment content under bare-ground conditions. It indicate that the bare land began to runoff and sediment production about 10 minutes late the rainfall, and the runoff generation time was slightly ahead of sediment yiled. Then, about 5.2 m$^3$/s and 140 kg/m$^3$ were the maximum flood-peak flow and the maximum sand content after 40 min and 45 min of rainfall, respectively. When the terraced fields were arranged in the upper, middle, and lower parts of the watershed, the runoff time was 45 min, 30 min, and 20 min late rainfall, and there were 35 min, 20 min, and 10 min later compared with the bare land, respectively. The nearer to the lower part the terraces were, the smaller the impact of the flow and sediment-production time. The maximum peak flow was 4.0 m$^3$/s, 4.1 m$^3$/s and 4.0 m$^3$/s, which were 77%, 8%, and 77% of the maximum flow in the bare land, respectively. The times of appearance were 115 min, 85 min, and 60 min after the rainfall, which were 2.5 times, 2.1 times, and 1.5 times later than that of the bare ground, respectively. The maximum sediment content was 110 kg/m$^3$, 100 kg/m$^3$, and 105 kg/m$^3$, which were 78%, 71% and 75% of the bare land, respectively. The sediment peaks times were 125 min, 130 min, and 135 min after the rainfall, which were 2.7 times, 2.9 times, and 3.0 times later than the bare land, respectively. It shows that there were time lags of runoff and sediment production, compared with bare land. The closer the terrace was to upstream, the greater the impact on the production of runoff and sediment.

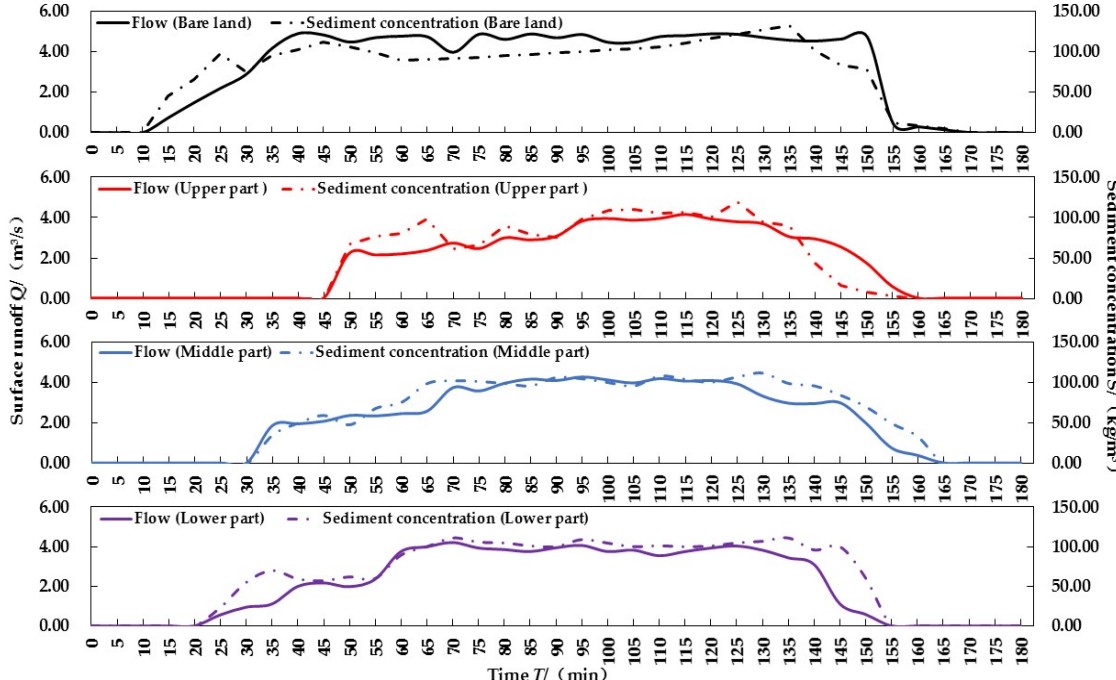

**Figure 4.** Comparison of water and sediment changes in different spatial positions (*y*-axis) of terraced fields in high-management watershed.

Figure 5 shows the relationships between the different parameters of runoff and sediment yield as well as the relative distance under the condition, where terraces account

for 30% of the watershed. A common feature was the existence of quadratic-function relationships between the relative distance ($r$) and time of runoff generation ($T_c$) and confluence ($T_h$), flood ($T_f$) and sediment ($T_s$) peak duration, flood ($Q_h$) and sediment peak ($S_f$) values, and soil loss ($E$). Where $r \approx 0.5{\sim}0.6$, there were maximum values of $T_c$, $T_f$, and $Ts$ in Figure 5a,b as well as minimum values of $T_f$, $Q_h$, $S_f$, and $E$. There were also similar relationships between the above hydraulic parameters and the vertical relative distance, as shown in Table 7. The above situation showed that terraces arranged in the middle and upper parts of the watershed might achieve better water and sediment-reduction effects.

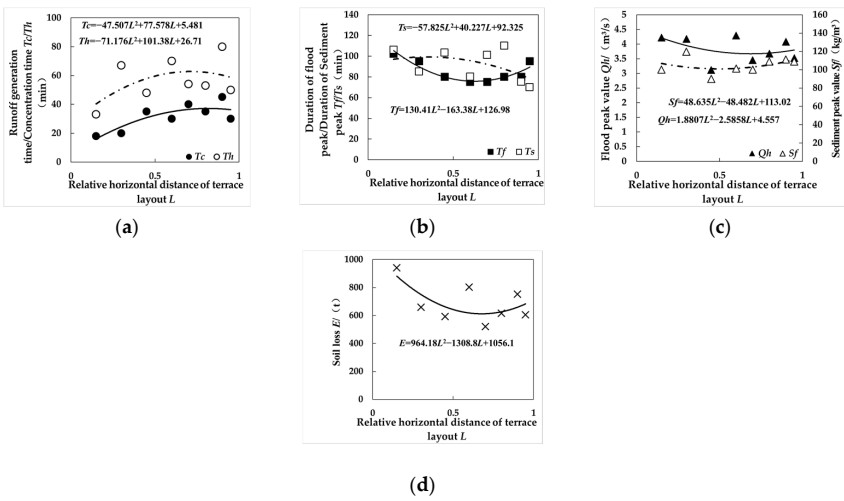

**Figure 5.** Variation law of erosion transport in watershed under different terrace longitudinal layout (*y*-axis). ((**a**) *L*-$T_c$/$T_h$; (**b**) *L*-$T_f$/$T_s$; (**c**) *L*-$Q_h$/$S_f$; (**d**) *L*-$E$).

**Table 7.** Variation law of erosion transport in watershed under different terrace vertical layout (*z*-axis).

| Direction | Parameter | Change Rule | Extremum | Note |
|---|---|---|---|---|
| Vertical (*z*-axis) | Runoff generation time | $T_c = -129.81Hr^2 + 120.09Hr + 10.607$ | 0.5 | Rainfall intensity: 1.14 mm/mim; Vegetation coverage: 70% |
| | Concentration time | $T_h = -125.93Hr^2 + 141.82Hr + 15.899$ | 0.5 | |
| | Duration of flood peak | $T_f = 221.24Hr^2 - 213.72Hr + 126.81$ | 0.5 | |
| | Duration of sediment peak | $T_s = -176.99Hr^2 + 118.98Hr + 88.238$ | 0.45 | |
| | Flood peak value | $Q_h = 3.5807Hr^2 - 3.8687Hr + 4.4265$ | 0.5 | |
| | Sediment peak value | $S_f = 64.587Hr^2 - 51.703Hr + 111.19$ | 0.45 | |
| | Soil loss | $E = 3353.4\,Hr^2 - 3370.3Hr + 1301.7$ | 0.5 | |

### *3.2. Effect of Spatial Variation of the Different Terrace Areas on Runoff and Sediment Yield in the Watershed*

#### 3.2.1. Effects of Different Terrace Areas at the Same Height on Watershed-Erosion-Transportation Processes

Five terrace areas were selected (10%, 20%, 30%, 40%, and 50%) to set up area gradients for rainfall experiments, compared with the erosion-transportation-processes under the condition of bare slopes. Figure 6 shows the variation laws of the hydraulic parameters with the relative area ($A$) of the terraced fields. The time of runoff yield ($T_c$) and confluence ($T_h$) were proportional to the terrace-layout relative area ($A$) in Figure 6a, while the duration of flood ($T_f$) and sediment peak ($T_s$) were inversely proportional to the terrace-layout area, and there were the quadratic function relationships between the terrace-relative area and flood peak ($Q_h$), sediment peak ($S_f$) value, and soil loss, respectively. When the ratio of terraces to the watershed area was 40%, the sediment peak value and soil loss reached the minimum, which were 103.8 kg/m$^3$ and 521.48 t, respectively. It showed that the layout of terraces should not exceed 40% of the watershed area.

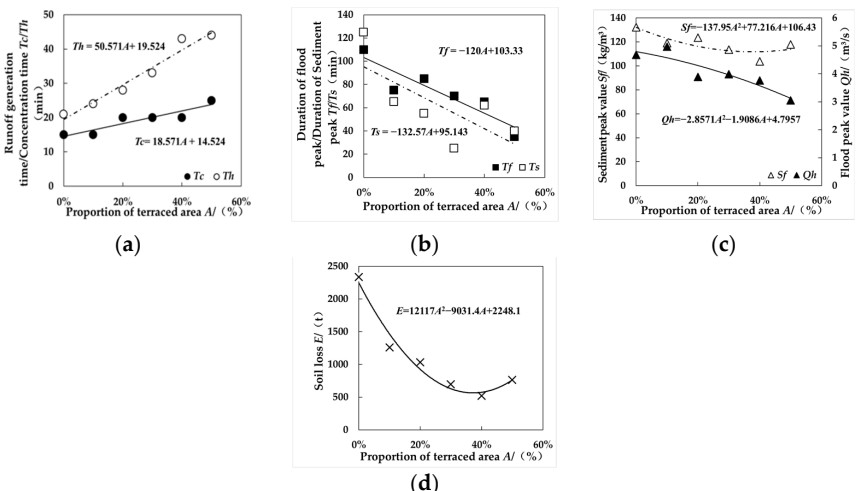

**Figure 6.** Variation law of watershed-erosion-transportation processes under the conditions of different terrace-laying areas (*x*, *y*-axis). ((**a**) $A$-$T_c$/$T_h$, (**b**) $A$-$T_f$/$T_s$, (**c**) $A$-$Q_h$/$S_f$, (**d**) $A$-$E$).

### 3.2.2. Effects of Different Terrace Areas and Relative Height on Watershed-Erosion-Transportation Processes

Figure 7 shows the variations of the hydraulic parameters with relative elevation (0.2, 0.25, 0.4, 0.55, and 0.8) of the relative areas of different terraces (10%, 20%, 40%, and 50%). Figure 7 shows that there were quadratic-function relationships between the relative height of terraces ($H_r$) and the time of runoff yield ($T_c$) as well as confluence ($T_h$) and duration of flood ($T_f$). Where $H_r \approx 0.4$~$0.6$, there were maximum values of $T_c$ and $T_h$ in Figure 7a,b, and minimum values of $T_f$ and $E$ in Figure 7c,d. Under the condition of the same height, the larger the layout area was, the longer the time of runoff yield and confluence time.

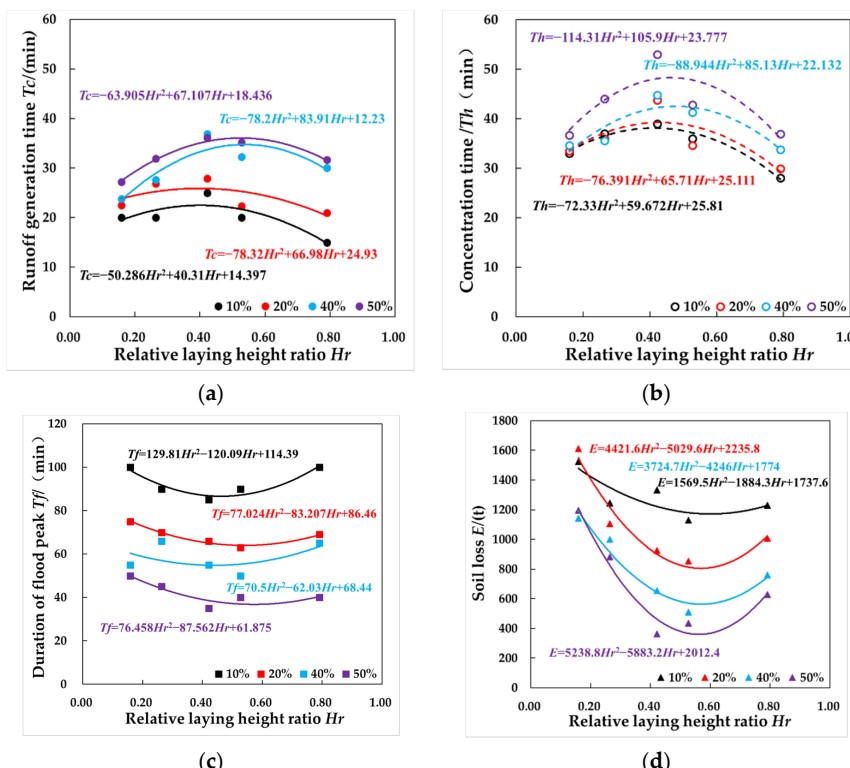

**Figure 7.** Variation law of watershed-erosion-transportation processes under the conditions of different terrace-laying areas and relative height. ((**a**) $H_r$-$T_c$, (**b**) $H_r$-$T_h$, (**c**) $H_r$-$T_f$, (**d**) $H_r$-$E$).

The duration of flood and soil loss also had a quadratic-function relationship with the relative-layout area and the height of terraces in the watershed. In addition, they had a minimal value around 0.4~0.6, which were 40~85 min and 400~1250 t, respectively. There was 77.67% erosion reduction of the terraces of the middle and upper parts, occupying 33% of the watershed area (Figure 7d). Under the condition of the same relative height, the smaller the layout area was, the greater the duration of the flood and the greater the soil loss.

### 3.3. Quantitative Evaluation of the Influence of the Spatial Distribution of Terraces on Watershed-Erosion-Transportation Processes

The above studies show that the location and area of terraces had important impacts for the runoff and sediment production. This effect should be expressed by the relationship between the erosion-reduction benefit of terraces and the relative position of terraces. If coordinate point o was used as the base point, as shown in Figure 2, and *r* was the ratio of the distance between the geometric center of the terrace and the longest distance ($L_{max}$) of the watershed, *r* was calculated using

$$r = \frac{L}{L_{max}} = \frac{\sqrt{x^2 + y^2 + z^2}}{\sqrt{x_{max}^2 + y_{max}^2 + z_{max}^2}} \tag{6}$$

Figure 8 is the relationship between the terrace-erosion-reduction benefit ($R_t$) and the space–distance ratio of terraces (*r*), based on the data of this experiment and other scholars (Table 8). Table 8 shows that the area of different watersheds ranges from 0.34 km² to 5000 km², the terrace area ranges from 0.1 km² to 300 km², and the erosion modulus varies from 5000 t/(a·km²) to 15,000 t/(a·km²). There was a nearly positive trend of the logarithmic function between $R_t$ and *r*, under the large-span watershed. When *r* was in the range of 0.1 to 0.35, $R_t$ increased the fastest. When *r* was about 0.35, the inflection point appeared. When *r* = 0.4~0.6, $R_t$ was close to the maximum. This showed that terraces arranged in the upper part of the watershed could receive better sediment-reduction benefits. Figure 8 was of great significance to the evaluation of sediment-reduction benefits and the design planning of terraces.

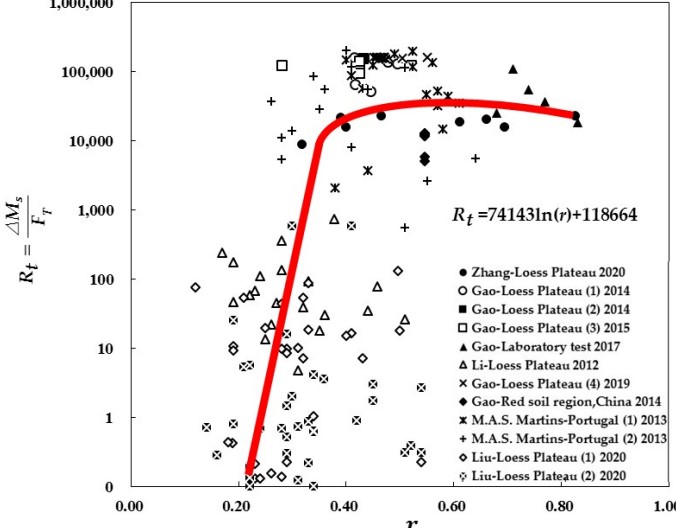

**Figure 8.** Relationship between the terrace-erosion-reduction benefit ($R_t$) and space–distance ratio of terraces (*r*).

**Table 8.** Information about different scholars.

| References | Geographical Situation | Area of Terraces (km$^2$) | Vegetation Coverage (%) | Relative Erosion Modulus t/(km$^2$·a) |
|---|---|---|---|---|
| Zhang and Gao | Yan River, Loess Plateau, China | 0.001~0.052 | 70~80 | 5000~8500 |
| Gao and Zhang [28] | Yan River, Loess Plateau, China | 0.002~0.007 | 60~65 | 1000~55,000 |
| Gao and Zhang [28] | Yan River, Loess Plateau, China | 0.002~0.004 | 60~65 | 1000~5000 |
| Gao and Bai [29] | Yan River, Loess Plateau, China | 0.0005~0.002 | 60~70 | 8500~32,000 |
| Gao and Wen [30] | Laboratory | 0.0000028 | — | 20~150 |
| Li and Gao [31] | Wuding River, Loess Plateau, China | 0.408~8.16 | 60~70 | 50~300 |
| Gao and Lin [32] | Yan River, Loess Plateau, China | 0.002~0.004 | 60~70 | 150~3500 |
| Shao andGao [15] | Red Soil Region, China | 0.001 | 70~80 | 300~5000 |
| M.A.S. Martins [33] | Granite Eucalyptus Slope, Portugal | 0.002~0.004 | 60~70 | 10~5000 |
| M.A.S. Martins [33] | Schist Pine Slope, Portugal | 0.002~0.004 | 60~70 | 5~1500 |
| Liu and Gao [21] | Zuli River, Loess Plateau, China | 500~5500 | 40~50 | 150~85,000 |
| Liu and Gao [21] | Weihe River, Loess Plateau, China | 500~25000 | 40~50 | 350~270,000 |

## 4. Discussion

### 4.1. The Test-Background Value and Result-Error Problem

This study was carried out under the 60–80% vegetation coverage of the Kangjiagelao watershed. However, to study the changes in the spatial location and area of terraces, it was necessary to eradicate part of the vegetation. In this way, it takes a long time to restore the vegetation to ensure that the model vegetation was similar to the prototype. This bought great difficulty to the completion of the model-test task. The way was to minimize vegetation eradication and to ensure similar vegetation coverage. This might have affected vegetation similarity, leading to an underestimation of the vegetation effect.

Another problem was the result error, as shown in Figure 8. Figure 8 indicated that the simulated test points were distributed around the trend line, but the verification data of other scholars were scattered. Since the watershed areas covered by the data were from 0.34 km$^2$ to 134,800 km$^2$, the terraced areas were from 0.0000028 km$^2$ to 25,000 km$^2$, the vegetation coverage ranges were from 30% to 80%, and the verification data of other scholars only roughly considered the coordinates of the center position of the terrace, and so ao; these possible factors contributed to the large variation in the results in Figure 8. In addition, the landscape pattern of terraces might also have a greater impact on the result values. This may also be an important question to study.

### 4.2. Further Innovations in Research Methods

The most common problem is the change and optimization of the spatial pattern of terraces, based on watershed erosion and transport processes. This is not only a current problem but also an important theoretical problem. After Shao proposed the terraced field module, 2008 SWAT488 was widely used to optimize the spatial pattern of terraced fields [15]. However, it is difficult to generalize, locate, and accurately verify for scattered terrace fields in the watershed. The watershed solid-scale model could, theoretically, test

the influence of each terraced field. This could make up for the problem of the long field-monitoring period and the difficulty in distinguishing the influencing factors. The relevant test results could also be used as the basis for validating the mathematical model. However, there were also some disadvantages, such as the long test period, especially the long vegetation-recovery time. This might affect the test progress. Consequently, it should be necessary to explore the theoretical and experimental methods of artificially simulating vegetation in the future.

*4.3. The Future Research Direction*

Previous studies have demonstrated that the research conclusions were different about the impact of terraced fields on water and sediment in the watershed under different study areas, scales, and geographical conditions [17]. The reasons might be related to the structure of the terraces, construction materials, years of operation, and vegetation coverage as well as types of vegetation in different study areas. Furthermore, the effects of terraces might also be different in different ecosystems, resulting in further changes in the service function of regional ecosystems [34,35]. In the future, it should be necessary to study the influence of terraced fields on the changes of water and sediment in the watershed under the conditions of different rainfall intensity, vegetation coverage, land-use type, and operating time. At the same time, it might also be necessary to select different study areas and multi-scale watershed-terrace projects to conduct experiments as well as to calibrate and verify the conclusions of this study.

**5. Conclusions**

Based on the above research and discussion, the following conclusions were drawn:

(1) The change of the spatial pattern of terraced fields in the watershed had important impacts on the processes of runoff and sediment. There was an approximate quadratic-function relationship between the spatial location and the parameters of runoff and confluence. When terraces were located the middle and upper parts ($r = 0.4$~$0.6$), there were extreme values of hydraulic parameters.

(2) The longitudinal distribution of $R_t$ was upper and middle > lower parts, and the vertical distribution of Rt was high > low place. The erosion reduction was 77.67% of the terraces of the middle and upper parts, occupying 33% of the watershed area. $R_t$ is the increasing logarithmic-function relationship with the center distance of the terraces ($r$). When $r$ is in the range of 0.1 to 0.35, $R_t$ increases the fastest. When $r$ is in the range of 0.35 to 0.45, the inflection point appears. When $r > 0.5$, $R_t$ grows slowly.

(3) The solid model of the three-dimensional scene reproduction of small watersheds had strong three-dimensionality and had great advantages in optimizing the structure as well as layout of soil- and water-conservation projects.

**Author Contributions:** Supervision, H.F., B.F., Y.Y., L.M., Q.J., A.L. and Y.Z.; writing—original draft, Z.G.; writing—review and editing, G.Z. and J.G. All authors have read and agreed to the published version of the manuscript.

**Funding:** This work was funded by the National Natural Science Foundation of China (Nos. 41877078 and 41371276), the National Key Research and Development Program of China (No. 2017YFC0504703), the Shaanxi Province Key Research and Development Program (No. 2020ZDLSF06-03), the Knowledge Innovation Program of the Chinese Academy of Sciences (No. A315021615), and the Shaanxi Province Science and Technology Innovation Project (No. 2013KTDZ03-03-01).

**Institutional Review Board Statement:** Not applicable.

**Informed Consent Statement:** Not applicable.

**Data Availability Statement:** Not applicable.

**Acknowledgments:** We acknowledge Anbin Li, Lin Ma, Boyan Feng, Yuanhao Yu, Ruolin Meng, Xinghua Li, Xingcheng Zhang, Saiqi Han, Shaobo Long, Fanfan Zhou, Sixuan Liu, Lu Wang, and Zhaorun Wang for their contributions to the indoor experiment. The careful reviews and constructive comments of the editors and anonymous reviewers are gratefully acknowledged.

**Conflicts of Interest:** The authors declare no conflict of interest.

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
