# Peer review of "Erosion-Transportation Processes Influenced by Spatial Distribution of Terraces in Watershed in the Loess Hilly–Gully Region (LHGR), China"

_water, doi:10.3390/w14121875_

Round 1

Reviewer 1 Report

Dear Authors

The manuscript titled "Erosion Transportation Processes as Influenced by Spatial Distribution of Terraces in Watershed in the Loess Hilly–Gully Region, China”

by Gao et al., is interesting. Please check all the detailed comments and questions provided below. Some are particular relevant to consider my recommendation for acceptance.

Detailed Comments

Introduction

There is no comparison between this topic and research in the world. I suggest to complete.

Materials and Methods

Was the initial moisture content measured before rainfall ?.

How was rain simulated?

Author Response

Dear Authors

The manuscript titled "Erosion Transportation Processes as Influenced by Spatial Distribution of Terraces in Watershed in the Loess Hilly–Gully Region, China” by Gao et al., is interesting. Please check all the detailed comments and questions provided below. Some are particular relevant to consider my recommendation for acceptance.

Detailed Comments

1.Introduction.There is no comparison between this topic and research in the world. I suggest to complete.

Response: Thank you very much for your comments. The introduction was partially revised (L40~83).

2.Materials and Methods. Was the initial moisture content measured before rainfall? How was rain simulated?

Response: Thank you very much for your comments. The materials and methods of this manuscript was partially revised (L118~123).

Reviewer 2 Report

The paper addresses important issues, both from the scientific and economic point of view, related to the planning/optimization of the terraces layout in terms of soil erosion control in the small watersheds. In order to determine the influence of the spatial arrangement of terraces on the amount of runoff and soil losses, the authors carried out a study on the physical model of the catchment area made on a scale of 1:100. The experiment was properly planned and executed, and the obtained results have application potential. A certain drawback of the work (quite significant) is the lack of a thorough analysis and discussion of the results obtained. The indication of further potential research directions may be indicated, but may not constitute the basis for discussion. In addition, the description of Figure 7 requires clarification. The figure shows that the maximum concentration of sediments was observed between 135-140 minutes, not, as the authors say, between 40 and 45 minutes (lines 207-208).  

In my opinion, the article requires correction and discussion improvements and after taking into account the reviewer's comments should be accepted for publication.

Some more specific comments 

  • Line 40 – what does the word "mu" mean?
  • Line 42 - the "and" should be removed at the end of the sentence.
  • Lines 52, 53 and 84 – the abbreviations should be clarified.
  • Line 61 - The citation is incorrect, Liu is not the first author of publications 20 and 21.
  • Table 4 „Meaning of Sign” should be "I" instead of "II".
  • When reading the text, I have a problem with understanding it, namely I do not know if the "model" and "prototype" (lines 99, 103, 105, table 6) refer to the same thing? This requires an explanation in the paper.
  • The differences between the description of table 6 (lines 180-182) and the information contained in it (no average velocity, maximum confluence) require explanation and clarification.
  • Whether the amount of rainfall capacity given in table 6 (140-150 mm)  is correct?
  • Lines 230, 232. Does the "r" symbol have the same meaning as the "L" symbol in Figure 5 (x-axis)? If so, I suggest that the markings be standardized throughout the article.

Author Response

The paper addresses important issues, both from the scientific and economic point of view, related to the planning/optimization of the terraces layout in terms of soil erosion control in the small watersheds. In order to determine the influence of the spatial arrangement of terraces on the amount of runoff and soil losses, the authors carried out a study on the physical model of the catchment area made on a scale of 1:100. The experiment was properly planned and executed, and the obtained results have application potential. A certain drawback of the work (quite significant) is the lack of a thorough analysis and discussion of the results obtained. The indication of further potential research directions may be indicated, but may not constitute the basis for discussion. In addition, the description of Figure 7 requires clarification. The figure shows that the maximum concentration of sediments was observed between 135-140 minutes, not, as the authors say, between 40 and 45 minutes (lines 207-208).

Response: Thank you very much for your comments. The issues were analyzed and discussed in L329~337.

Some more specific comments

1.Line 40 – what does the word "mu" mean?

Response: Thank you very much for your comments. The word “mu” is a municipal land area unit in China. 1 mu is equal to sixty square feet, or about 666.7m2.

2.Line 42 - the "and" should be removed at the end of the sentence.

Response: Thank you very much for your comments. This word has been deleted.

3.Lines 52, 53 and 84 – the abbreviations should be clarified.

Response: Thank you very much for your comments. These abbreviations have been clarified (L4,57~58).

4.Line 61 - The citation is incorrect, Liu is not the first author of publications 20 and 21.

Response: Thank you very much for your comments. The relevant literature has been remarked in the corresponding place in the paper (L66~67).

5.Table 4 Meaning of Sign” should be "I" instead of "II"

Response: Thank you very much for your comments. The original text has been modified accordingly (L156).

6.When reading the text, I have a problem with understanding it, namely I do not know if the "model" and "prototype" (lines 99, 103, 105, table 6) refer to the same thing? This requires an explanation in the paper.

Response: Thank you very much for your comments. The prototype refers to the Kangjiagelao small watershed introduced in Figure 1 and L93~100. The model (as shown in Figure 2) was built by scale conversion of the prototype according to table 5.

7.The differences between the description of table 6 (lines 180-182) and the information contained in it (no average velocity, maximum confluence) require explanation and clarification.

Response: Thank you very much for your comments. The average flow rate and maximum confluence have been added to table 6 (L201).

8.Whether the amount of rainfall capacity given in table 6 (140-150 mm) is correct?

Response: Thank you very much for your comments. The 140~150mm is the erosion equivalent rainfall of prototype. And relevant data have been converted into the prototype data (L193~200).

9.Lines 230, 232. Does the "r" symbol have the same meaning as the "L" symbol in Figure 5 (x-axis)? If so, I suggest that the markings be standardized throughout the article.

Response: Thank you very much for your comments. The r and L have different meanings. The L was the distance from the geometric center of the terrace to the origin (L237~241), and the r is the ratio of the distance between the geometric center of the terrace and the longest distance (Lmax) of the watershed (L302~L304).

Reviewer 3 Report

The authors have done a tremendous amount of fundamental work to update knowledge about runoff on terraced slopes.
The work has been done at a level that allows us to assess the validity and reproduce (if desired) the experiment.
I recommend the manuscript with corrections to the text.
A separate request is for a comparison of the values obtained with existing erosion models. Also interesting is the inter-terrace runoff, what is the contribution from the cascade of terraces to the runoff in the lower part of the slope?

Author Response

The authors have done a tremendous amount of fundamental work to update knowledge about runoff on terraced slopes.

The work has been done at a level that allows us to assess the validity and reproduce (if desired) the experiment.

I recommend the manuscript with corrections to the text. 我推荐有正文更正的手稿。

A separate request is for a comparison of the values obtained with existing erosion models. Also interesting is the inter-terrace runoff, what is the contribution from the cascade of terraces to the runoff in the lower part of the slope?

Response: Thank you very much for your comments. L338~L350 was comparison between the results of this study and existing models. The Figure 4 indicated that runoff interception by cascaded terraces reduced the catchment of runoff under the slope.
